# Dose Intensive Rituximab and High-Dose Methylprednisolone in Elderly or Unfit Patients with Relapsed Chronic Lymphocytic Leukemia

**DOI:** 10.3390/medicina55110719

**Published:** 2019-10-29

**Authors:** Regina Pileckyte, Vilma Valceckiene, Mindaugas Stoskus, Reda Matuzeviciene, Jurgita Sejoniene, Tadas Zvirblis, Laimonas Griskevicius

**Affiliations:** 1Hematology, Oncology and Transfusion Medicine Center, Vilnius University Hospital Santaros Klinikos, 08661 Vilnius, Lithuania; vilma.valceckiene@santa.lt (V.V.); mindaugas.stoskus@santa.lt (M.S.); tadas.zvirblis@gmail.com (T.Z.); laimonas.griskevicius@santa.lt (L.G.); 2Institute of Clinical Medicine, Vilnius University, 08661 Vilnius, Lithuania; 3Laboratory Diagnostics Centre, Vilnius University Hospital Santaros Klinikos, 08661 Vilnius, Lithuania; reda.matuzeviciene@santa.lt; 4Radiology and Nuclear Medicine Center, Vilnius University Hospital Santaros Klinikos, 08661 Vilnius, Lithuania; jurgita.sejoniene@santa.lt

**Keywords:** chronic lymphocytic leukemia, rituximab, high-dose methylprednisolone, elderly

## Abstract

*Background and Objectives*: BTK and BCL2 inhibitors have changed the treatment paradigms of high-risk and elderly patients with chronic lymphocytic leukemia (CLL), but their long-term efficacy and toxicity are still unknown and the costs are considerable. Our previous data showed that Rituximab (Rtx) and high-dose methylprednisolone (HDMP) can be an effective and safe treatment option for relapsed high-risk CLL patients. *Materials and Methods*: We explored the efficacy and safety of a higher Rtx dose in combination with a shorter (3-day) schedule of HDMP in relapsed elderly or unfit CLL patients. *Results*: Twenty-five patients were included in the phase-two, single-arm trial. The median progression free survival (PFS) was 11 months (range 10–12). Median OS was 68 (range 47–89) months. Adverse events (AE) were mainly grade I–II° (77%) and no deaths occurred during the treatment period. *Conclusions*: 3-day HDMP and Rtx was associated with clinically meaningful improvement in most patients. The median PFS in 3-day and 5-day HDMP studies was similar and the toxicity of the 3-day HDMP schedule proved to be lower. The HDMP and Rtx combination can still be applied in some relapsed high-risk and elderly or unfit CLL patients if new targeted therapies are contraindicated or unavailable. (ClinicalTrials.gov identifier: NCT01576588).

## 1. Introduction

Chronic lymphocytic leukemia (CLL) is the most prevalent type of leukemia among adults [1]. The majority of patients are older than 70 and have coexisting conditions.

Before the era of targeted therapy, standard treatments included combinations of purine analogues, alkylating agents, and monoclonal antibodies. Chemoimmunotherapy can provide long-term disease control in younger patients without major comorbidities, but may have unacceptable side effects in older patients in whom non-myelotoxic approaches may be preferred.

New agents such as BTK and BCL2 inhibitors have changed the treatment paradigms of high-risk and/or elderly patients with CLL [1,2], but the long-term efficacies and toxicities are still unknown and the costs are considerable.

Rituximab (Rtx) and high-dose methylprednisolone (HDMP) can be an effective and safe treatment option for relapsed CLL in different settings [3,4,5]. Earlier studies [4,5] have explored the combination of HDMP with the anti-CD20 antibody Rtx in relapsed and fludarabine-refractory CLL patients using various Rtx dosing schedules and treatment durations. Our previous study of HDMP and Rtx combination in relapsed high-risk CLL, including *TP53* dysfunction, demonstrated efficacy with an overall response rate (ORR) of 62% [3]. However, there were three deaths throughout the study. In order to reduce toxicity, we have attempted to evaluate the combination of a shorter HDMP schedule and an Rtx dose that is higher than the standard dosage. Here, we present the results of the HDMP and Rtx combination given to elderly, pretreated patients.

## 2. Materials and Methods

### 2.1. Patients

A phase II, single-arm, open-label, prospective study was conducted to evaluate HDMP in combination with Rtx in elderly or unfit patients with relapsed CLL.

Eligible patients had CD20+ CLL with treatment indications according to the criteria of the International Workshop on Chronic Lymphocytic Leukemia (IWCLL 2008) [6]. They had relapsed or progressive disease symptoms after at least one prior chemotherapy regimen, and were 18–64 years of age with comorbidities (cumulative illness rating scale (CIRS) >6) or ≥65 years of age irrespective of comorbidities. Exclusion criteria included intolerance to exogenous protein administration or previous reaction to Rtx treatment, active infections, tuberculosis or fungal infections within the past six months, active peptic ulcers, severe organ deficiency preventing participation in the study, inadequately controlled diabetes mellitus, allergic disorders in need of chronic glucocorticoid therapy, and pregnancy or lactation.

### 2.2. Assessments and End Points

Pretreatment evaluation consisted of laboratory tests, a bone marrow biopsy and aspiration with immunophenotyping, and a neck, chest, abdominal, and pelvic computer tomography (CT). The coding region of the *TP53* gene (exons 2–11) was PCR-amplified and scanned for mutations by high-resolution fluorescent melting (HRM) curve analysis as previously described [7]. Mutations were confirmed by direct sequencing. Immunoglobulin heavy-chain variable region (*IgHV*) mutational status was determined by direct DNA sequencing [8]. The sequences were aligned to *IgHV* sequences from the IMGT database. Gene sequences deviating more than 2% from the corresponding germline gene were defined as mutated. The presence of 17p deletion, 11q deletion, trisomy 12, and 13q deletion was assessed by FISH analysis using commercially available probes (Kreatech Diagnostics, Amsterdam, The Netherlands). ZAP70 expression in microbeads purified CD19+ cells (purity over 99%) was analyzed using RT-qPCR. The cut-off value for ZAP70 expression was determined using CD19+ cells from 30 healthy donors. CIRS assessment was performed according to guidelines [9]. Adverse events were reported according to the National Cancer Institute Common Toxicity Criteria (version 4.0) [10], and hematological toxicity was evaluated according to IWCLL 2008 guidelines [6].

The response to therapy was confirmed by CT two to three months following the end of treatment, and complete responses were confirmed by bone marrow biopsy. Minimal residual disease (MRD) was analyzed according to international guidelines [6]. During the follow-up period, patients underwent physical and laboratory tests every three months until disease progression ceased or death occurred.

The primary objective was to determine the treatment response rate. Secondary objectives were used to determine progression-free survival (PFS), overall survival (OS), and the safety profile of Rtx and HDMP. The study protocol was approved by the Lithuanian Bioethics Committee and the study was conducted according to the Declaration of Helsinki. The ethical code number is P-11-005 and the date of approval was 13-01-2011. All patients provided written informed consent.

### 2.3. Study Treatment and Monitoring

HDMP was administered at a daily dose of 1 g/m^2^ intravenously over 4 h for three consecutive days for four cycles. After 14 patients had been included, the protocol was amended to allow an additional two HDMP cycles to be given to patients without significant toxicity. Rtx was administered at a dose of 1000 mg/m^2^, following HDMP infusion for 4 courses. To decrease the incidence of initial infusion reactions, patients received the first dose of Rtx split into 50 mg on day one, 150 mg on day two, and the remaining 800 mg on day three. A full dose was given on the first day during 2–4 courses and the regimen was repeated every 21 days. There were no dose adjustments for Rtx. If non-hematological clinically significant grade III–IV toxicities related to glucocorticoid occurred, the HDMP dose could be decreased by 50% during subsequent doses.

### 2.4. Statistical Methods

Statistical analysis of survival rates and responses according to IWCLL 2008 guidelines were performed on an intent-to-treat basis for all enrolled patients. Adverse events (AEs) and clinical safety data were summarized using descriptive statistics. Response to treatment was expressed as the proportion of patients who achieved at least a partial response (PR). Paired Student’s t-test was used to compare blood count values during treatment. Survival trends were evaluated using the Kaplan–Meier method. Cox regression analysis was used to evaluate the impact of different prognostic factors on PFS and OS. Two-tailed *p* values <0.05 were considered significant. Statistical analysis was performed using SAS version 9.2. RP, TZ, and LG analyzed the data. All authors had access to primary clinical trial data.

### 2.5. Comparison of LT-CLL-001 and LT-CLL-2s Study Results

Indirect comparison of treatment schedule, prognostic factors, PFS, OS, and salvage therapy were described.

## 3. Results

### 3.1. Patient Characteristics

Between October 2011 and October 2014, 27 patients were screened, with 25 patients being included in the study. Two patients were screen failures: one due to active hepatitis B and another because of concomitant myelodysplastic syndrome. Patient demographics and baseline characteristics are summarized in Table 1. The median age was 73 years (range 65–80), the CIRS median was five (range 1–8), patients had received a median of two (1–6) previous treatment lines, 80% of the patients suffered from B symptoms, five (20%) had 17p deletion and/or *TP53* mutation, three (12%) had 11q deletion, 17 (68%) had unmutated *IgVH*, 16 (64%) had hyperexpression of ZAP70, and 96% had elevated β_2_ microglobulin >3.5 mg/L.

Sixteen patients were given ≤4 courses (one stopped treatment after two courses due to disease progression, the remaining patients received the planned four courses). After protocol amendment, nine patients received >4 courses (eight patients were given six planned courses, and one patient received five courses due to an adverse event).

### 3.2. Response and Survival

Overall response rate was 28% and all were partial responses, mainly due to residual lymphadenopathy. No MRD negative cases were confirmed. All 20 patients with B symptoms had their symptoms resolved. Significant improvement in anemia (*p* < 0.001) and reduction of lymphocytosis (*p* < 0.001) were noted (Figure 1). No significant change in neutrophil and platelet counts were observed.

The median follow-up for all patients was 50 (11–74) months, and 57 (range 41–74) months for the 12 patients who are still alive. The PFS was 11 months (range 10–12) (Figure 2). Median OS was 68 (range 47–89) months (Figure 3). No differences in response, PFS (*p* = 0.888), and OS (*p* = 0.152) were noted between the patients who received ≤4 or >4 treatment courses. Three-year PFS was 5% and OS was 64%.

Twelve (44%) patients were alive at the last evaluation. Two patients failed to attend follow up and another 11 died. All patients progressed, but only one patient without indication for salvage treatment. One patient died due to complicated cholecystitis one year after study treatment completion and prior to salvage therapy. Other causes of death included disease progression in three patients, pre-existing cardiovascular disorders in two patients, and infections after salvage treatment in four patients. The cause of one patient’s death is unknown.

### 3.3. Prognostic Factors for PFS and OS

Gender, bulky lymphadenopathy, 17p deletion and/or *TP53* mutation, *IgVH* mutational status, number of study treatment courses, ZAP mutations, CD 38 expression, elevated beta-2 microglobulin, and CD20 expression were evaluated in Cox regression analysis for PFS and OS (Table 2). Univariate analysis showed that 17p deletion/*TP53* mutation, bulky lymphadenopathy, positive ZAP70 expression, unmutated *IgVH*, and increased β_2_ microglobulin were predictive of shorter PFS. High CD20 expression (>90%) was predictive of longer PFS in univariate analysis. Multivariate analysis confirmed the presence of 17p deletion/*TP53* mutation, unmutated *IgVH*, and high β_2_ microglobulin as adverse factors for PFS. There was a trend of shorter OS in patients with 17p deletion/*TP53* mutation and higher β_2_ microglobulin.

### 3.4. Toxicity

Clinically significant adverse events were recorded during treatment and up to three months after the last treatment cycle (Table 3). AEs were mainly grade I–II (81%) and consisted primarily of cardiovascular events (hypertension, supraventricular extrasystolia, atrial fibrillation) and hypokalemia and infections (upper respiratory tract infections, urine tract infection, herpes labialis). Hypokalemia was transient, related to HDMP administration, and was corrected by potassium supplements without clinically significant relevant sequelae. Hematologic toxicity was limited and consisted primarily of neutropenia. Seven patients experienced 16 episodes of grade III–IV neutropenia. Three cases of febrile neutropenia were observed. G-CSF was administered in five (20%) patients. There were no deaths during the treatment period. Grade III–IV infections were noted in three patients (12%). The HDMP dose was not reduced for any patient.

### 3.5. Salvage Therapy

Twenty-two patients received salvage treatment after the study. The median time to first treatment was 19 (3–46) months. Eight (36%) of 22 LT-CLL-2s patients received ibrutinib as salvage treatment. Median (range) OS from first post-study relapse treatment was 17 (4–30) months in chemo (immuno) therapy and 44 (41–47) months in the ibrutinib group (*p* = 0.085). There was a trend of better survival for patients given ibrutinib at any point during disease progression (Figure 4).

## 4. Discussion

Before the era of pathway inhibitors, monoclonal anti-CD20 antibodies and high-dose glucocorticoid combinations were evaluated in poor-prognosis patients (Table 4).

Castro et al. [4] treated 14 refractory or fludarabine-intolerant patients with HDMP-Rtx and achieved an ORR of 93% and a CR of 36% with favorable safety profiles. In this study, the highest cumulative Rtx dose was applied, however, only three patients had a poor prognosis and CT scans were not included in the response evaluation.

In our previous study [3], we evaluated a Rtx/HDMP combination for relapsed high-risk patients. Higher HDMP dose per treatment course (5 g/m^2^ vs. 3 g/m^2^) and slightly lower Rtx dose per complete treatment period (3375 mg/m^2^ vs. 4000 mg/m^2^) was used in the LT-CLL-001 study compared to the LT-CLL-2s study (Table 4). The median age was 59 years old vs. 73 years old, and there were more high-risk patients in LT-CLL-001 vs. LT-CLL-2s: *TP53* mut/17p del 44% vs. 20%, *IgHV* unmutated, 86% vs. 68%, fludarabine refractory 34% vs. 8%. The ORR was 62% (all PRs) vs. 28% (all PRs), PFS was 12 vs. 11 months, OS was 31 vs. 68 months, median follow-up was 31 vs. 50 months, and there were three vs. zero treatment-related deaths in the LT-CLL-001 vs. LT-CLL-2s study, respectively. Nine patients (31%) were ≥65 years of age in the LT-CLL-001 trial and seven (78%) responded to this treatment. The median PFS in this patient group was 13 months and the median OS was 40 months. There was one death during the treatment period of an elderly patient due to gastrointestinal bleeding.

Eight (36%) of LT-CLL-2s patients received ibrutinib vs. one (5%) of 20 LT-CLL-001 patients as salvage, which could have contributed to longer OS in the LT-CLL-2s study.

Simkovic et al. [11] presented retrospective data of Rtx and high-dose dexamethasone combination with an ORR of 75%, median PFS of 8 months, and median OS of 25.5 months. ORR was not dependent on age, but in multivariate analysis, an age of ≥65 years and an absence of therapeutic response (SD/PD) were identified as independent predictors of shorter PFS (*p* = 0.002 for both). Only an age of ≥65 years was a significant predictor of shorter OS (*p* = 0.006), which may be explained by higher treatment toxicity in the elderly population.

Doubek et al. [12] evaluated 33 relapsed refractory CLL patients treated with at least three cycles of an ofatumumab and high-dose dexamethasone combination. ORR was 67%, the median PFS was 10 months, and the median OS was 34 months.

Smolej et al. [13] ran the largest prospective study, combining high-dose dexamethasone with two schedules of high-dose Rtx (1500 mg/m^2^ on days 1, 8, 15, 22 with 375 mg/m^2^ in the first dose, repeated every 4 weeks (group I) and 2500 mg/m^2^ on day one (375 mg/m^2^ in first cycle) repeated every 3 weeks (group II)), evaluated 54 relapsed or refractory (R/R) CLL patients with ORR of 62–72% and a high CR rate of 21% in patients receiving a lower dose of Rtx but also having a lower incidence of bulky disease. Response evaluation did not include bone marrow examination, and imaging methods included CT scans or ultrasound examination. Median follow-up was less than 12 months with a median PFS of 6–9 months and a median OS of 14 months (group I) vs. not reached (group II). Serious infections occurred in 32% of the patients, resulting in three early deaths.

Overall, the ORR was more than 60% in these studies and the median PFS was 8–12 months. The ORR in our study was only 28%, largely due to residual lymphadenopathy confirmed by CT imaging. However, our patients achieved the median PFS of 11 months, which is comparable to the previous studies. In the LT-CLL-001 study, the interim analysis at three months and after the end of therapy [3] demonstrated increasing hemoglobin and platelet levels, as well as a 17% improvement in lymph node response. These observations suggest that longer exposure to high doses of gliucocorticoids translates into a better response and may influence progression-free survival but may also cause higher toxicity.

Grade III–IV infections were diagnosed in 21–27% of cases and the treatment-related mortality was 10–12% in glucocorticoid/Rtx combination studies (Table 4). Notably, grade III–IV infections were also observed in 24%, and 4% of patients died in the Ibrutinib arm of the RESONATE trial [2] (Table 4).

None of the previous trials were specifically designed for the elderly, relapsed patient population which is known to be more fragile and susceptible to infectious complications. In our study, III–IV° infections were observed in 12% of patients. Importantly, there were no treatment-related deaths (Table 4).

The longer median OS in the LT-CLL-2s study (68 months) compared to the LT-CLL001 study (31 months) may have been due to the lower number of high-risk patients (*TP53* mut/17p del 20% vs. 44%, *IgHV* unmutated, 68% vs. 86%, fludarabine refractory 8% vs. 34%), the higher availability for BTK inhibitors as salvage therapy (36% vs. 5% of patients), and a significantly lower toxicity (treatment related mortality 0% vs. 10%) in the LT-CLL-2s study. This was achieved despite the LT-CLL-2s patients being older than those in the LT-CLL-001 study (median age 73 vs. 59 years, respectively).

## 5. Conclusions

In conclusion, when targeted therapies are unavailable or contraindicated, a HDMP and Rtx combination may be applied either as a 5-day HDMP regimen in relapsed refractory high-risk cases or a 3-day regimen in elderly or unfit CLL patients. Further studies of HDMP and Rtx combination in CLL refractory to pathway inhibitors are warranted.

## Figures and Tables

**Figure 1 medicina-55-00719-f001:**
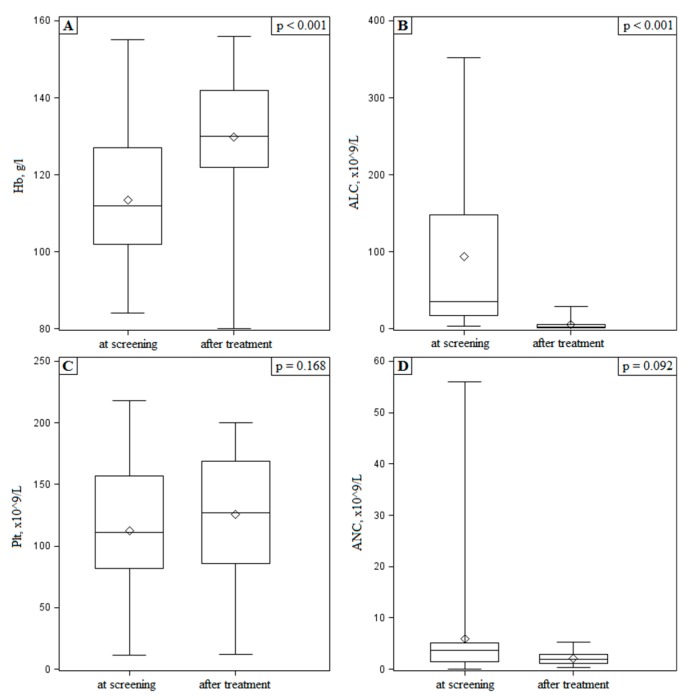
(**A**)—Hemoglobin (Hb); (**B**)—absolute lymphocyte count (ALC); (**C**)—platelet (Plt); (**D**)—absolute neutrophil count (ANC) at screening and cessation of treatment.

**Figure 2 medicina-55-00719-f002:**
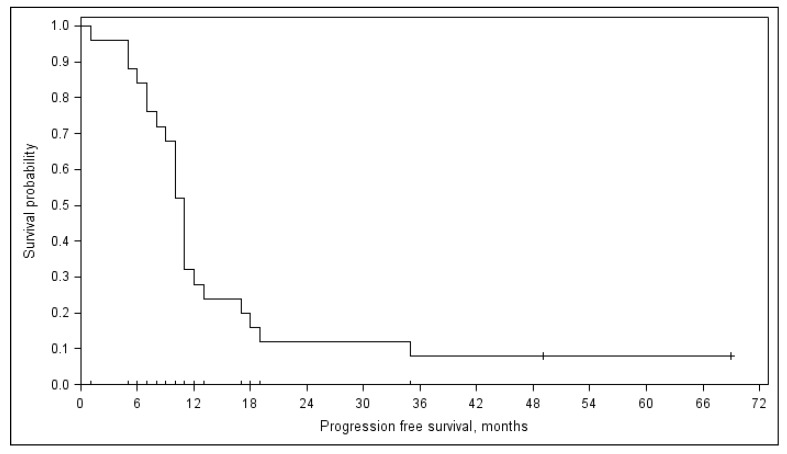
Progression-free survival (PFS).

**Figure 3 medicina-55-00719-f003:**
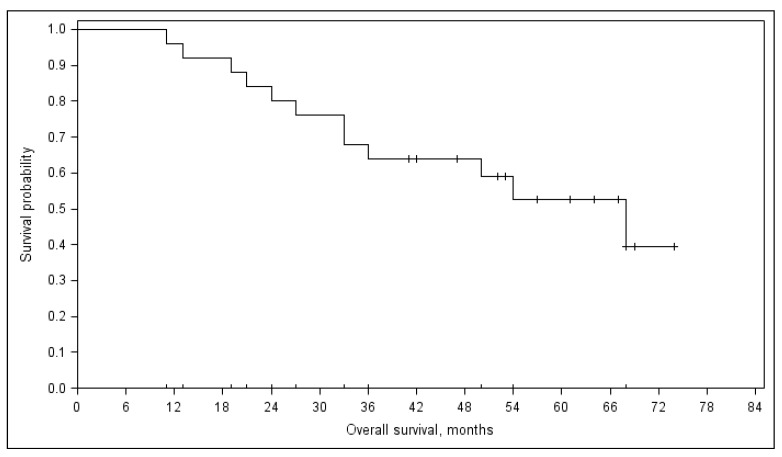
Overall survival (OS).

**Figure 4 medicina-55-00719-f004:**
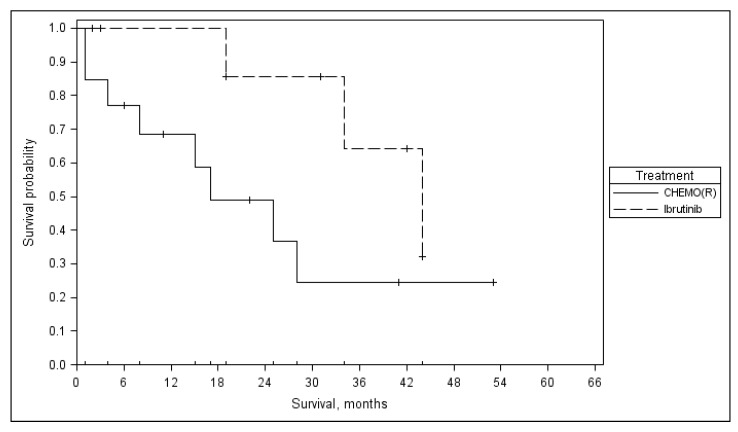
Overall survival from first salvage according to given treatment.

**Table 1 medicina-55-00719-t001:** Patient baseline characteristics.

Characteristic	N° of Patients	%
Number of patients	25	100
Median age, years (range) ≥75 years	73 (65–80) 9	36
Gender		
Male	7	28
Female	18	72
CIRS median (range) Median creatinine clearance (mL/min)	5 (1–8) 54	
Rai stage		
I–II	10	40
III–IV	15	60
Bulky lymphadenopathy (>5 cm)	12	48
B symptoms	20	80
CD20 expression >80%	18	72
B2 mikroglobulin >3.5 mg/L *	23	96
ZAP 70 possitive	16	64
*IgHV* unmutated	17	68
Genetics		
17p deletion and/or *TP53* deletion ** 11q deletion only *** Trisomy 12 only 13q deletion only	5 3 2 7	20 12 8 28
Median no. of prior treatments (range) Median time from the last treatment, months (range)	2 (1–3) 20 (0–153)	
Fludarabine containing regimen Rituximab containing regimen	7 3	28 10

* 24 patients tested for B2 microglobulin ** One patient had 11q and 13q gene deletion, and one additional patient had a trisomy 12 deletion. *** One patient also had 13q gene deletion.

**Table 2 medicina-55-00719-t002:** Univariate Cox regression analysis of impact of different prognostic factors on PFS and OS.

Characteristics	PFS			OS		
	HR	95% CI	*p*	HR	95% CI	*p*
Male gender	0.840	0.319–2.208	0.723	1.117	0.333–3.750	0.858
p17del/*TP53* mutation *	6.314	1.905–20.930	0.003	3.285	0.952–11.337	0.060
Cycles No ≤4 vs. >4	1.060	0.448–2.507	0.895	0.344	0.074–1.598	0.174
Bulky disease > 5 cm	2.847	1.140–7.114	0.025	2.081	0.656–6.607	0.214
ZAP expression	3.850	1.343–11.039	0.012	1.392	0.418–4.641	0.590
*IgVH* mutation *	0.168	0.051–0.548	0.003	0.319	0.070–1.466	0.142
CD38 expression ≥ 20%	1.070	0.469–2.440	0.872	0.595	0.189–1.877	0.376
CD20 expression > 90%	0.976	0.956–0.996	0.019	0.987	0.960–1.014	0.332
B_2_ microglobulin > 3.5 mg/L *	1.303	1.045–1.625	0.019	1.228	0.999–1.510	0.051

* Independent predictive factors for PFS in multivariate Cox regression.

**Table 3 medicina-55-00719-t003:** Adverse events.

Adverse Events	Grade I/II n (%)	Grade III/IV n (%)	Subjects with at Least One AE n
All reported	106 (81)	25 (19)	xxx
Chills	1 (9.4)		1
Cardiovascular disturbances	17 (16.0)	1 (4.0)	15
Gastrointestinal disturbances	4 (3.8)		3
Hematology		4 (16.0)	4
Neutropenia	2 (1.9)	* 16 (64.0)	7
Infections	11 (10.4)	4 (16.0)	8
Urticaria papulosa	1 (9.4)		1
Lumbalgia subacuta	3 (2.8)		2
Arthralgias/headache	4 (3.8)		4
Cough	3 (2.8)		3
Hyperuricemia	1 (9.4)		1
Renal insufficiency	2 (1.9)		2
Thrombosis	2 (1.9)		2
Fever	5 (4.7)		3
Edema	5 (4.7)		4
Left femur cervical stress fracture	1 (9.4)		1
Cognitive disturbance	1 (9.4)		1
Hyperglycemia	10 (9.4)	1 (4.0)	2
Hypokalemia	30 (28.3)		18
Insomnia	3 (2.8)		3

* Three cases of febrile neutropenia.

**Table 4 medicina-55-00719-t004:** Summary of high-dose glucocorticoid and antiCD20 antibodies combination treatment in the literature.

Reference	Pts No	Age, Median	No of HDMP or DEXA Cycles (Days)	Total Dose of Rtx mg/m^2^	Pt with 17pdel No. (%)	ORR/CR (%)	Median PFS, Months	Median OS, Months	AE III–V°
Castro et al. [4]	14	62	3 (5)	5250	1 (7)	93/36 *	15	NR **	One case of pneumonia
Simkovic et al. ***** [11]	60	66	8 (8)	3875	11 (23)	75/3	8	25, 5	27% serious infections Treatment related mortality 10%,
Doubek et al. [12]	33	66	3/6 (8)		8 (24)	67/15	10	34	III–IV° infections 21%, Treatment related mortality 12%
Pileckyte et al. [1]	29	59	6 (5)	3375	13 (44)	62/0	12	31	III–IV° infections 23% Treatment related mortality 10%
Pileckyte et al.	25	73	4/6 (3)	4000	5 (20)	28/0	11	68	III–IV° infections 12% No treatment related deaths
Byrd et al. [2] Ibrutinib	195	67			63 (32)	43/0	At median follow-up of 9,4 months, median PFS not reached	At 12 months, the overall survival rate was 90%	≥III° infections 24% Treatment related mortality 4%

* CT scans not performed for response evaluation. ** Not reached after a median follow up of 40 months. *** Retrospective study.

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
