# Peer review of "Dose Intensive Rituximab and High-Dose Methylprednisolone in Elderly or Unfit Patients with Relapsed Chronic Lymphocytic Leukemia"

_medicina, 2019, doi:10.3390/medicina55110719_

Round 1

Reviewer 1 Report

In the current manuscript, the authors studied efficacy and safety of higher rituximab dose in combination with shorter (3-day) schedule of HDMP in relapsed elderly or unfit CLL patients. They report on response, survival rate and adverse effects. The work is interesting and presents valuable data for the benefit of patients with this disease.
I suggest that the authors consider the comments below:

The lines and pages refer to the word document:

The title, please use Relapsed Chronic Lymphocytic Leukemia instead of  Chronic B Lymphocytic Leukemia

Text, page 1, line 34: from below: „Chemoimmunotherapy can provide
long-term disease control in younger patients without major comorbidities but may have unacceptable side effects in older patients in whom non-myelotoxic approaches may be preferred.“ Authors should mention that frontline G-Clb treatment could provide acceptable long-term results in elderly patient with CLL. I recommend focussing on relapsed or refractory patients only.

Text, page 2: Methods: Were patients hospitalized for the administration of HDMP?

Text, page 2, line 57: from below: previous reaction to Rtx treatment – What does it mean reaction? Is it infusion related reaction (IRR)? How was this exclusion defined? It would not be fair to exclude pts with grade 1 IRR from rituximab treatment in subsequent treatment line.

Text, page 2, line 79: MRD testing – could you specify the assay for MRD testing. Bone marrow or peripheral blood was tested? What was the threshold for minimal residual disease?

Text, page 2, line 90: additional two HDMP cycles in patients without significant toxicity. What was the definition of significant toxicity?

Text, page 3, lines 102-104: The definition of PFS and OS interval is redundant.

Text, page 3, line 116: What is the explanation for high prevalence of B symptoms?

Text, page 4, line 121: Patient baseline characteristics. What is the explanation for high proportion of females in this study, as CLL is more prevalent in men?

Text, page 1, line 57:

Text, page 4, line 121: Patient baseline characteristics -Fludarabine containing regimen – Does it mean: Pretreated with fludarabine regimen?

Text, page 4, line 121: Patient baseline characteristics – Genetics and IgHV: What proportion of patients were tested? Do we have any data on karyotype?

Text, page 4, line 121: Patient baseline characteristics – Information on creatinine clearance should be added to the table. This is important comorbidity.

Genetics and IgHV: What proportion of patients were tested? Do we have any data on karyotype?

Text, page 4, line 121: Why is reported CD20 expression >80%? In Cox regression analysis the cut-off > 90 % is used.

Text, page 5, line 141: What method has been used for the calculation of median follow up?

Text, page 7, line 177: G-CSF was used only as secondary prophylaxis of neutropenia?

Text, page 7, line 178: Please list all infections.

Text, page 7, line 179: Table – Adverse event. Can you specify Cardiovascular disturbancies

And Gastrointestinal disturbancies in note below table. Consider renaming these AEs or use disturbances instead of disturbancies.

Text, page 9, line 190: Patient outcome in LT-CLL-001 compared to LT-CLL-2s study – This is interesting part of the manuscript. In the first place, the comparison of these two studies should be also mentioned in the methods. Secondly, more details on statistical results should be enclosed (esp. p values).

Text, page 9, line 198: “which could have contributed to longer OS in the LT-CLL-2“ – this should be included in Discussion.

Text, page 9-10, Table IV. The references are not listed correctly. I recommend not the include the study Byrd et al. – apple and oranges are compared here.

Text, page 9-10, Table IV. Study published by Smolej et al. (Leu Res, 2012) should also incorporated in the table, as this is one of the biggest serios on combination of rituximab and high-dose corticosteroids in R/R CLL pts. Moreover, the comparison of ORRs with the current study will be feasible in the subsequent text.

Text, page 11, line 227: The ORR in our study was only 28%, mostly due to residual lymphadenopathy. – I am not sure, that this relevant explanation of low ORR rate in this study. Do you see any other reason for that?

Text, page 11, line 233-236: Please compare the median follow up of patients in these studies.

Text, page 11, line 244-245: This was achieved despite LT-CLL-2s patients were older than in LT-CLL-001 study (median age 73 vs 59 years, respectively). Was the difference of statistical significance?

Reviewer 2 Report

1) Please specify how faste was relapse after last prior regiment? Line 55. And put this information in Table I.

2) How did you define “inadequately controlled diabetes”? Line 59.

3) Which were considered toxicities related to glucocorticoid? Line 95. In lines 173-174 you state that “hypokaliemia was transiet related to HDMP”, but what about connection HDMP and infections?

4) In line 96 - “HDMP dose could be decreased by 50% during subsequent doses”. But I did not find data was it done or not? Please specify this and write in line 178.

5) In 53-54 lines you write “treatment indications according … IW CLL 2008” which includes also “spleen palpable more than 6 cm below costal margin”. Please specify spleen size and put it in Table I.

6) Table I –Bulky lymphadenopathy ( > 5 cm). But according to IW CLL 2008 “massive nodes > 10 cm in longest diameter”. Please specify this criteria for “bulky” ?

7) Table I. About prior treatment – please specify how many patients recieved Rituximab containing regiments?

8) Please specify when was the last evaluation (line 152) – how many months after treatment?

9) 5. to be deleted from the article at all, as it does not refer to this study and its results.

10) 5.1. please include in discussion, lines 226-232.
